# Sensorimotor synchronization to music reduces pain

**Lucy M. Werner**[1,2¤], **Stavros Skouras**[1], **Laura Bechtold**[2], **Ståle Pallesen**[3], **Stefan Koelsch**[1]*

1 Department for Biological and Medical Psychology, University of Bergen, Bergen, Norway, 2 Department of Biological Psychology, Institute for Experimental Psychology, Heinrich Heine University, Düsseldorf, Germany, 3 Department of Psychosocial Science, University of Bergen, Bergen, Norway

¤ Current address: Institute of Clinical Neuroscience and Medical Psychology, Medical Faculty, Heinrich Heine University, Düsseldorf, Germany
* Stefan.Koelsch@uib.no

**Data Availability Statement:** All relevant data are within the paper and its Supporting information files.

**Funding:** The current project was funded by Grant 260576 from the Research Council of Norway awarded to S.K., and the Trond Mohn Stiftelse

## Abstract

Pain-reducing effects of music listening are well-established, but the effects are small and their clinical relevance questionable. Recent theoretical advances, however, have proposed that synchronizing to music, such as clapping, tapping or dancing, has evolutionarily important social effects that are associated with activation of the endogenous opioid system (which supports both analgesia and social bonding). Thus, active sensorimotor synchronization to music could have stronger analgesic effects than simply listening to music. In this study, we show that sensorimotor synchronization to music significantly amplifies the pain-reducing effects of music listening. Using pressure algometry to the fingernails, pain stimuli were delivered to $n = 59$ healthy adults either during music listening or silence, while either performing an active tapping task or a passive control task. Compared to silence without tapping, music with tapping (but not simply listening to music) reduced pain with a large, clinically significant, effect size ($d = 0.93$). Simply tapping without music did not elicit such an effect. Our analyses indicate that both attentional and emotional mechanisms drive the pain-reducing effects of sensorimotor synchronization to music, and that tapping to music in addition to merely listening to music may enhance pain-reducing effects in both clinical contexts and everyday life. The study was registered as a clinical trial at ClinicalTrials.gov (registration number NCT05267795), and the trial was first posted on 04/03/2022.

## Introduction

Pain-reducing effects of music listening are among the most widely studied and most replicable effects of music in clinical settings [1, 2]. For example, a recent meta-analysis by Kuhlmann et al. [2], with 7385 patients in 92 randomized controlled trials, reported that music listening interventions significantly reduce pain in surgical patients. However, across four different meta-analyses, the pain-reducing effects of music listening had only small to moderate effect sizes (and were inconsistent among the studies analyzed), which calls the clinical relevance of music for pain relief into question [1–4]. Notably, almost all of the studies included in the aforementioned meta-analyses used passive listening interventions (the study by Lee [3]

(TMS) / Bergens Forskningsstiftelse (BFS). The funders had no role in study design, data collection and analysis, decision to publish, or preparation of the manuscript.

**Competing interests:** The authors have declared that no competing interests exist.

reports 10 music therapy studies of which six were using an active intervention such as group singing [5–10]), and to the best of our knowledge no study has compared pain-reducing effects between passive music listening and active engagement in music.

The lack of research comparing pain-reducing effects between these two forms of music listening (i.e active vs. passive) is noteworthy because recent advances have proposed that active sensorimotor synchronization to music is associated with the activation of the endogenous opioid system (EOS), including the release of endorphins [11, 12]. This notion is supported by previous studies that had already associated music making with activation of the EOS [13–15]. Notably, EOS activation supports both analgesia and social bonding [12]. Hence, recent evolutionary accounts have posited that the human capacity to synchronize movements to a musical pulse in a group (such as group singing, clapping, drumming, dancing) is an evolutionarily adaption because it stimulates social bonding [11, 12, 16–18]. However, to the best of our knowledge, the hypothesis of pain reduction by sensorimotor synchronization to music has never been investigated. Since pain-reduction is commonly used as a proxy of EOS activation [11–13, 16–20], sensorimotor synchronization to music, if it indeed activates the EOS, should have larger pain-reducing effects than simply listening to music.

Details on the study sample are shown in Fig 1.

The experimental design is illustrated in Fig 2 (for details see Materials and methods). Participants rated their perceived pain, emotional state (felt pleasantness and arousal), and familiarity with and preference for music during a 2x2 within-subjects design experiment where pain was applied to their fingernails using pressure algometry [21–23] while either listening to music or undergoing a silent control period and either performing an active foot tapping task or a passive control task. Emotion ratings were obtained to explore whether the mechanisms driving pain-reducing effects of sensorimotor synchronization to music include emotion (based on our previous work on music-evoked emotions [24] and previous research showing that modulatory effects of music on pain were mediated by the pleasantness of the emotions induced [25]. Familiarity and preference ratings were obtained to elucidate possible contributions of these factors on pain reduction [26] and to measure additional aspects with emotional implications.

In accordance with the well-established pain-reducing effects of music listening in clinical settings [1–4], we expected perceived pain to be reduced while listening to music compared with the silent control condition (independent of the task, i.e. independent of whether participants were tapping or not). Furthermore, consistent with the well-established effects of attention on pain [27–30], we expected pain-reducing effects of active tapping compared with no tapping (independent of the condition, i.e., independent of whether music or silence was presented), because active tapping requires increased attention on the part of the participants and thus distracts from perceived pain. Most importantly, we investigated whether participants felt less pain while actively tapping to music compared to passively listening to music, which would reveal, for the first time, analgesic effects of sensorimotor synchronization to music.

## Materials and methods

### Participants

A random sample of 59 participants, who were naïve with regard to the hypotheses, was included in the data analysis (age range 19 to 35 years, $M$ = 22.15 years, $SD$ = 3.22, 29 females). This sample size was based on an a-priori sample size estimation using G*Power (version 3.1.9.6) [31] which indicated that for the detection of differences between listening and not listening to music with an α-level of 0.05 (one-tailed) and a statistical power of 90% assuming a small to medium effect size of Cohen's $d$ = 0.4 [32] based on Cepeda et al. [1] and Lee [3], a

## CONSORT 2010 Flow Diagram

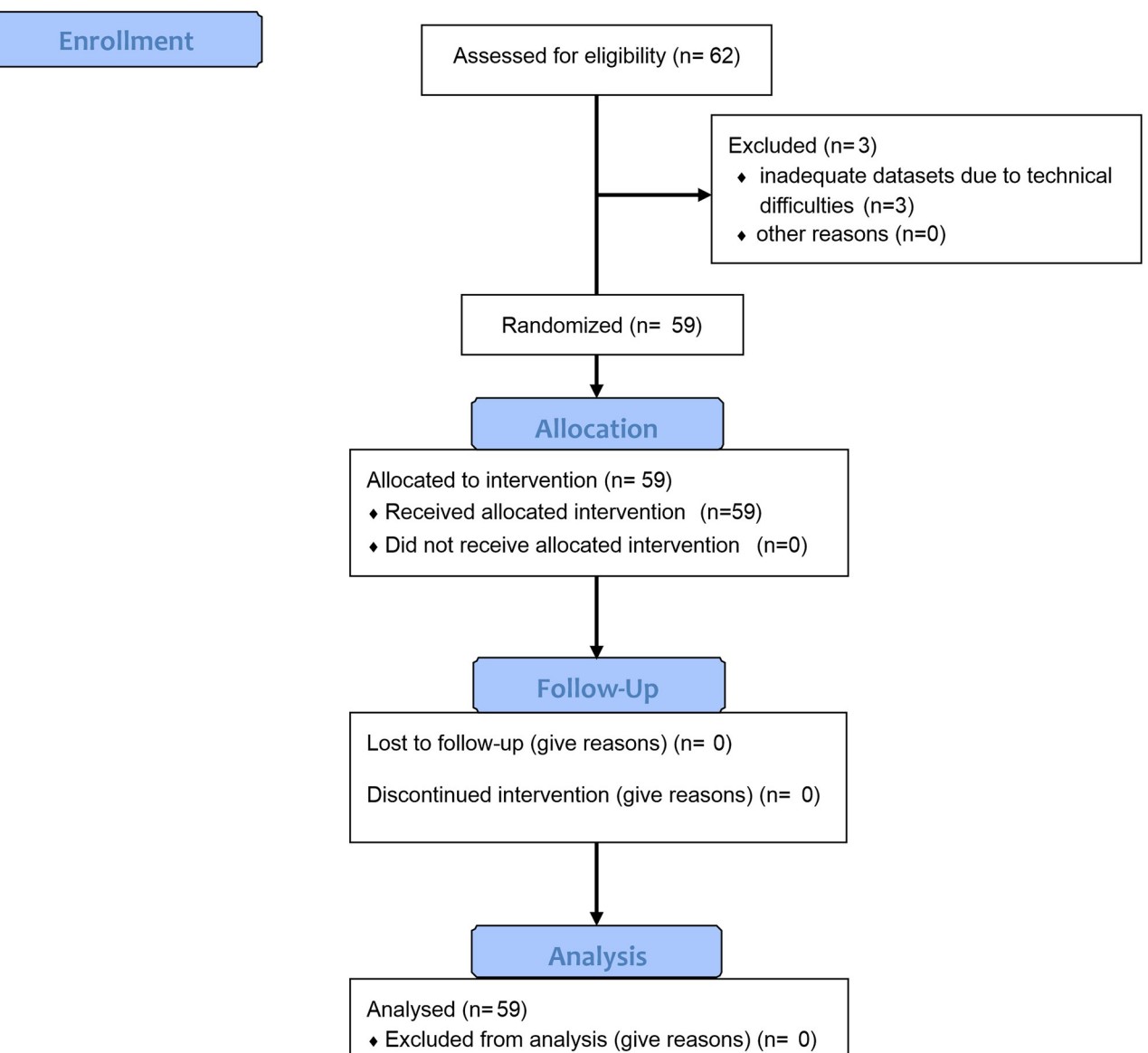

**Fig 1. Consort flow diagram on the study sample.**

sample of $n = 59$ is adequate. Exclusion criteria were use of any prescription drugs, psychiatric or neurological disease, hearing impairment, and history of substance dependence (according to self-report). Participants did not consume alcohol nor any medicine for the treatment of pain at least 24 hours prior to the experiment. None of the participants had musical anhedonia according to the Barcelona Music Reward Questionnaire ($M = 81.06$, $SD = 11.37$) [33]. $n = 53$ participants were right-handed, $n = 4$ were left-handed, and $n = 2$ were ambidextrous (EHI

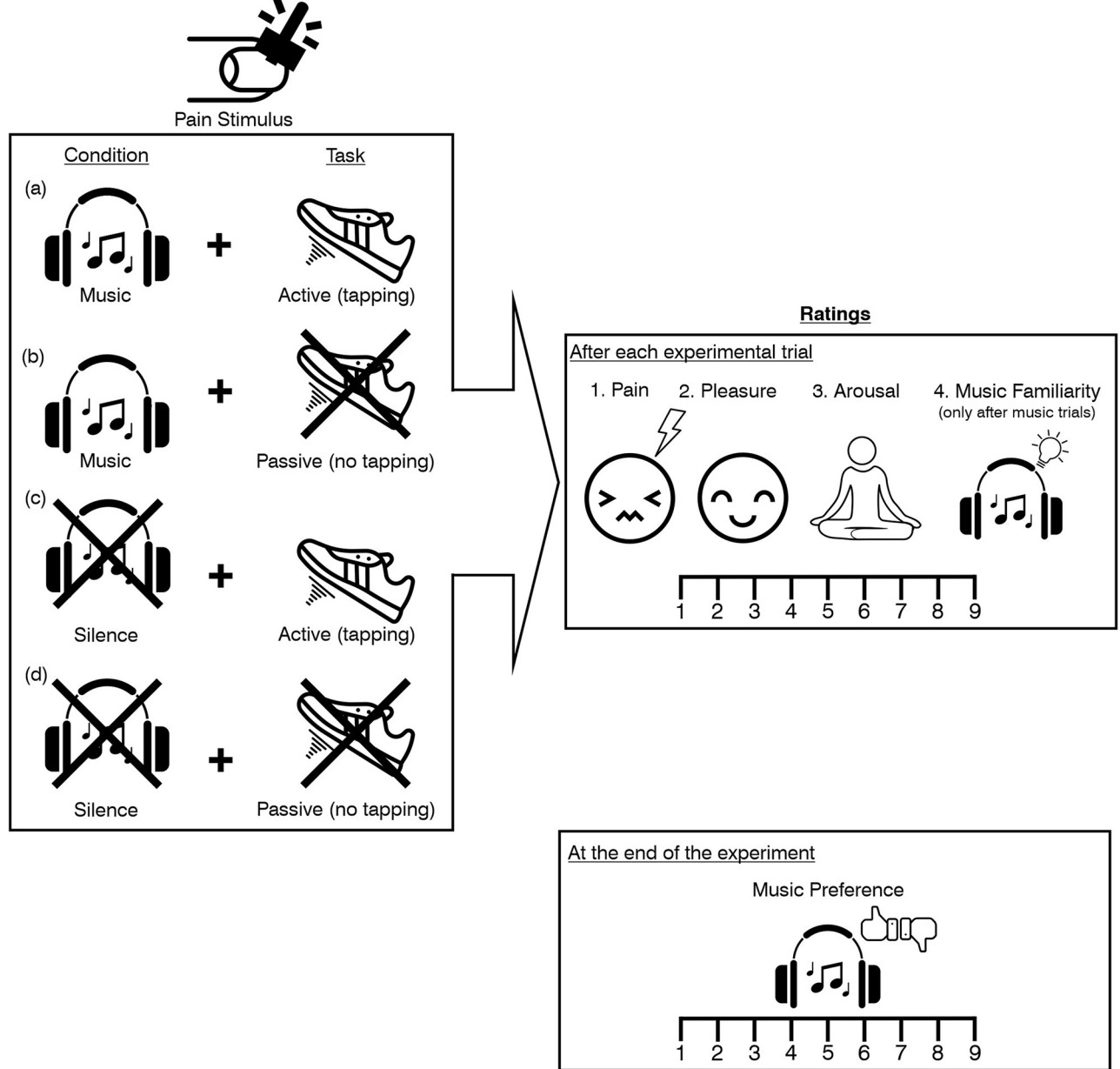

**Fig 2. Experimental design.** The experiment used a 2x2 design with the within-subject factors *Condition* (music, silence) and *Task* (active, passive), resulting in four experimental trial types: **(a)** *Music Active* (music with tapping); **(b)** *Music Passive* (music without tapping); **(c)** *Silence Active* (silence with tapping); and **(d)** *Silence Passive* (silence without tapping). The allocation of the music excerpts to the task (active, passive) was random, and the order of the four experimental trial types was counterbalanced. Specific pain levels were applied on the participants' fingernails in each of 40 experimental trials using pressure algometry. At the end of each trial (after the presentation of a music excerpt or after a silent period), participants rated (1) their perceived pain, (2) their emotional state with regard to felt pleasantness and (3) felt arousal, as well as (4) their familiarity with the music excerpt (only during trials with music). All ratings were provided on a scale ranging from 1 to 9. After the 40 experimental trials, participants provided preference ratings for each musical excerpt (also using a scale ranging from 1 to 9).

from -1.00 to 1.00, $M = 0.64$, $SD = 0.44$) [34]. For further details on the study sample see Fig 1. The study was carried out in accordance with the guidelines of the Declaration of Helsinki and approved by the Regional Committee for Medical and Health Research Ethics for Western Norway (Reference Number: 2019/1031). The authors confirm that all ongoing and related

trials for this intervention are registered. The study was registered as a clinical trial at Clinical-Trials.gov (registration number NCT05267795) upon request of the journal, and the trial was first posted on 04/03/2022. The trial protocol and the supporting CONSORT checklist are available as supporting information (see S1 Protocol and S1 Checklist). All participants provided written informed consent before enrollment and received a monetary compensation of 200 NOK following participation in the experiment.

## Stimuli

Music stimuli were presented with an average sound pressure of approximately 60 dB SPL over Beyerdynamic DT 770-PRO 250 Ohm headphones. 10 instrumental music experts were selected, each 30 seconds long (song characteristics are provided in S1 Table). Each excerpt was played twice during the experimental task, once during the active task and once during the passive task. Stimuli were delivered using the Matlab-based toolbox Cogent 2000 (version 1.33).

## Pressure algometry

Pain was applied by the experimenter during each trial using a Wagner Force One Pressure Algometer (Wagner Instruments, Greenwich, USA). The equipment head (used to apply the pressure) had a diameter of 12mm and a surface area of 113mm$^2$. Pain stimuli were delivered to index, middle and ring fingers of both hands. For each participant, pain thresholds were determined separately for each of the six fingers used for pain delivery, prior to the actual experiment. The average value of these pain thresholds was then computed, and for the actual experiment, only 50% of the participant's average pain threshold was delivered as pain stimulus in each of the 40 experimental trials (e.g., if a participant had an average pain threshold of 5 kg, then a pressure stimulus of 2.5 kg was applied in the actual experiment). During the entire experiment, the applied pressure was recorded on a computer with a 10 Hz sampling rate using MESURgauge Plus by Mark-10 (version 2.0.5). Repeated measures ANOVAs of these data showed that neither the applied pressure, nor the duration of applied pressure, differed significantly for the within-subjects factors *Condition* (music, silence; pressure intensity: $F(1,58) = 0.33$, $p = .568$; pressure duration: $F(1,58) = 1.62$, $p = .209$), nor *Task* (active, passive; pressure intensity: $F(1,58) = 0.33$, $p = .568$; pressure duration: $F(1,58) = 0.66$, $p = .419$). Likewise, no significant interaction between *Condition* and *Task* was observed for the pressure intensity ($F(1,58) = 1.00$, $p = .321$), nor for pressure duration ($F(1,58) < 0.01$, $p > .999$). Thus, we can exclude the possibility that the pain ratings of participants were simply an artifact of faulty stimulus delivery (note that the experimenter was also blinded to avoid any systematic differences of pain stimulus delivery between the experimental trial types). For further information see S1 File and S2 Table.

## Procedure

The investigation took place in a laboratory room at the department of biological and medical psychology of the University of Bergen. 40 experimental trials were delivered, 10 for each experimental trial type (Music Active, Music Passive, Silence Active, Silence Passive). The experimenter was blinded during the entire experimental procedure. Each trial started with an instruction screen where participants were either instructed to tap their right foot like a metronome (active tapping task) or to relax (passive control task). For exact experimental instructions see S2 File. Then, participants pressed the space bar and either music started to play, or a silent period started, and participants either had to tap or relax (depending on the trial type), while looking at a fixation cross in the middle of the screen. After 20 seconds the experimenter

applied pressure with the Algometer to a fingernail for 10 seconds. Then (after the pain application had stopped), participants indicated the pain they felt at the end of the pain application on a 9-point scale (1 = *very little*, 5 = *medium*, 9 = *very strong*), followed by a pleasantness rating (1 = *very uncomfortable*, 5 = *medium*, 9 = *very comfortable*) and an arousal rating (1 = *very calm*, 5 = *medium*, 9 = *very activated*). In music trials, participants also rated to which extent they were familiar with the music (1 = *not at all*, 5 = *partially known*, 9 = *well known*). Participants were aware that the experimenter could not see their ratings (so that they would not feel influenced in any way). Each trial lasted about one minute (thus, the experiment had a duration of approximately 40 minutes). After completing all 40 experimental trials, participants listened to the 10 music excerpts again and indicated their preference for each excerpt on a 9-point scale (1 = *strongly disliked*, 5 = *medium*, 9 = *strongly liked*).

### Experimental design & data analysis

Data analysis was computed using R (version 4.0.4) including the R Stats, R Base, and ggplot 2 (version 3.3.3) [35] packages. Figures were graphically edited with CorelDraw Graphics Suite version 21.0.0.593 (Corel Corporation, Ottawa, Ontario, Canada). We conducted an LME analysis on single trial level, as this analysis allowed to reduce interindividual variance introduced by participants and music excerpts, and decrease the risk for false positives [36]. The analysis was applied by the use of the lme4package (version 1.1–26) [37]. The model included the two categorical fixed-effects factors *Condition* (music [+0.5] and silence [-0.5]) and *Task* (active [+0.5] and passive [-0.5]) as well as their interaction as predictors for the perceived pain in each individual experimental trial (rated on a scale ranging from 1 to 9). Furthermore, the model included the Participants and music Excerpts as random-effects factors. The model was estimated by using a restricted maximum likelihood approach [38]. The α-level for significance was 0.05. We estimated degrees of freedom and *p*-values with the Satterthwaite approximation implemented in the lmerTest package (version 3.1) [39]. We used the R package influence.ME (version 0.9–9) [40] to test the data for statistical outliers on the participant level. No outliers were detected, and the model was conducted with a sample of 59 participants. To comprehensively explore the pattern of results, we conducted a simple slope LME analysis implemented in the R package jtool (version 2.1.2) [41] with *Task* as predictor and *Condition* as moderator. We calculated Cohen's *d* effect size [32] by the use of the R package lsr (version 0.5) [42] because this effect size measure is taken as one of the most important indicators of clinical significance [43]. In addition, we calculated (one-tailed) paired-samples t-tests between all four experimental conditions by the use of the R Stats package to further emphasize the clinical relevance of our study. The α-level for significance was 0.05.

### Results

Fig 3 shows the mean data of the perceived pain, dependent on *Condition* (listening to music, or silent control condition), and *Task* (performing the active tapping task, or the passive control task). A linear mixed effects (LME) analysis indicated significant main effects of both *Condition* ($\beta$ = -0.72, *SE* = 0.20, *p* = .027), and *Task* ($\beta$ = -0.20, *SE* = 0.07, *p* = .005), reflecting that the perceived pain was reduced while listening to music (compared to silence; see red vs. blue violin plots in Fig 3), and while performing the active tapping task (compared with the passive control task; see darker vs. lighter colors in Fig 3). For all inferential statistics see S3 Table. Importantly, the pain-reducing effect of the active tapping task was descriptively larger while listening to music (*mean difference* = 0.28 scale points, *SE* = 0.08, *d* = 0.46) than during silence (*mean difference* = 0.12 scale points, *SE* = 0.09, *d* = 0.17). A paired-samples t-test supports this

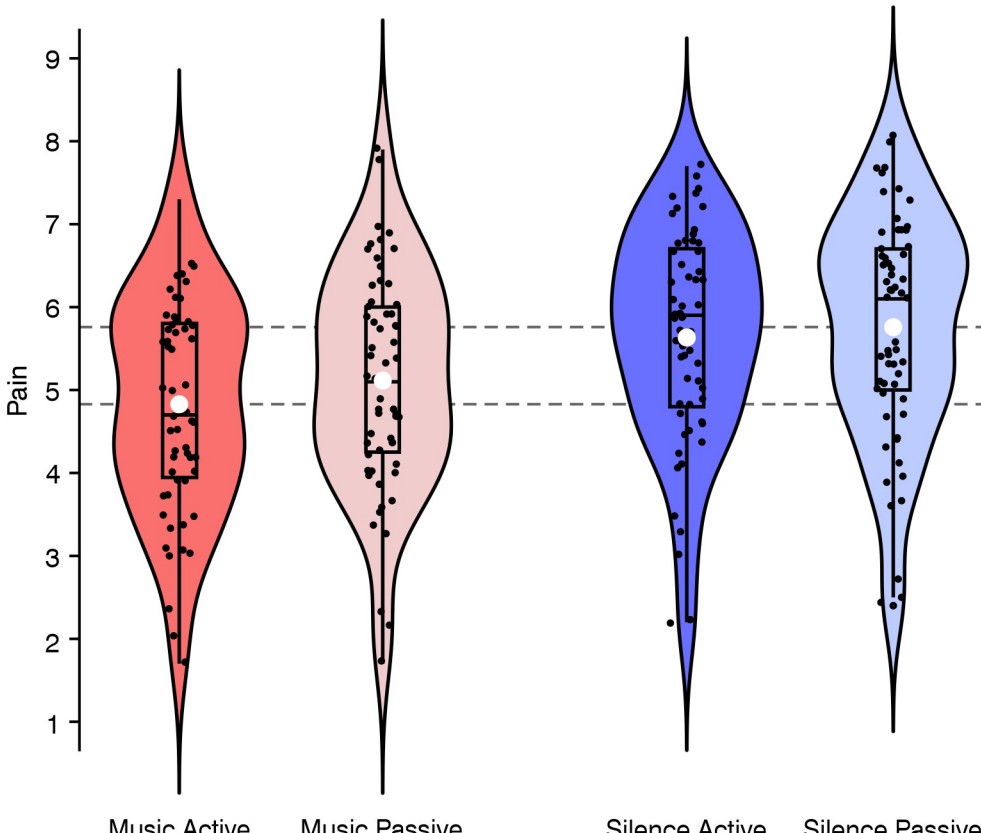

**Fig 3. Within-subjects perceived pain per experimental condition.** The violin plots show the distribution density of the perceived pain (rated on a scale ranging from 1 to 9) during music (red) or silence (blue) while performing either an active tapping task (darker colors) or a passive control task (lighter colors). The embedded box-and-whisker plots represent the 25th and the 75th percentiles of the distributions, respectively. Upper and lower whiskers extend from the hinge to the largest/smallest value no further than 1.5 * inter-quartile range. The vertical lines in the boxes indicate median values, and the white disks indicate the means. The black dots show the jittered data points, and the dashed grey horizontal lines in the background represent the mean difference of the perceived pain between music with tapping and silence without tapping. Note here that the pain-reducing effect of music with tapping (Music Active, dark red) compared with silence without tapping (Silence Passive, light blue) has a large effect size ($d = 0.93$).

strong pain-reducing effect of tapping to music compared to tapping in silence, $t(58) = -3.50$, one-sided $p < .001$.

Despite the observation of the strongest pain-reducing effect of tapping to music in the descriptive analysis, the LME analysis failed to reveal a significant interaction. However, it is important to note that the absence of a significant interaction does not necessarily imply that the tapping was not effective. From a clinical perspective, the absence of statistically significant differences must be carefully scrutinized [43, 44]. Given the clinical relevance of our effects, and the fact that the results pattern nominally followed our a priori hypotheses, we separately examined and statistically compared the conditions. On that account, we conducted a simple slope analysis with *Task* as predictor and *Condition* as moderator. This simple slope analysis revealed a significant effect of the *Task* only while listening to music ($\beta = -0.29$, $SE = 0.10$, $p = .005$) but not during silence ($\beta = -0.12$, $SE = 0.10$, $p = .225$). Consequently, the difference in perceived pain was most pronounced between music with tapping vs. silence without tapping *(mean difference* = 0.93 scale points, $SE = 0.13$, see dashed grey

horizontal lines in Fig 3), with a large effect size of *d* = 0.93 (effect sizes are interpreted according to Cohen's convention [32]). This effect was also confirmed by a paired-samples t-test, $t(58)$ = -7.17, one-sided $p < .001$ (for paired-samples t-tests on the perceived pain between all four conditions see S4 Table).

## Emotional mechanisms underlying the pain-reducing effect of sensorimotor synchronization to music

To test our assumption that the mechanisms driving the pain-reducing effect of sensorimotor synchronization to music include emotion, we computed two additional LME analyses. For detailed information on the experimental design and data analysis for both analyses see S3 File. The first included only music trials and the factors *Task*, *Preference* and *Familiarity*. The dependent variable was perceived pain. Fig 4 shows the preference effect on the perceived pain for the music trials only, separately for the active tapping task and the passive control condition. Consistent with the first analysis reported above, our second analysis also revealed a significant main effect of *Task* ($\beta$ = -0.28, *SE* = 0.10, $p$ = .006) with reduced perceived pain for the active vs. passive task. Importantly, this analysis further revealed a significant main effect of *Preference* ($\beta$ = -0.14, *SE* = 0.03, $p < .001$), indicating that pain ratings decreased with increasing preference for the music excerpts. The interaction of the factors *Task* and *Preference* was

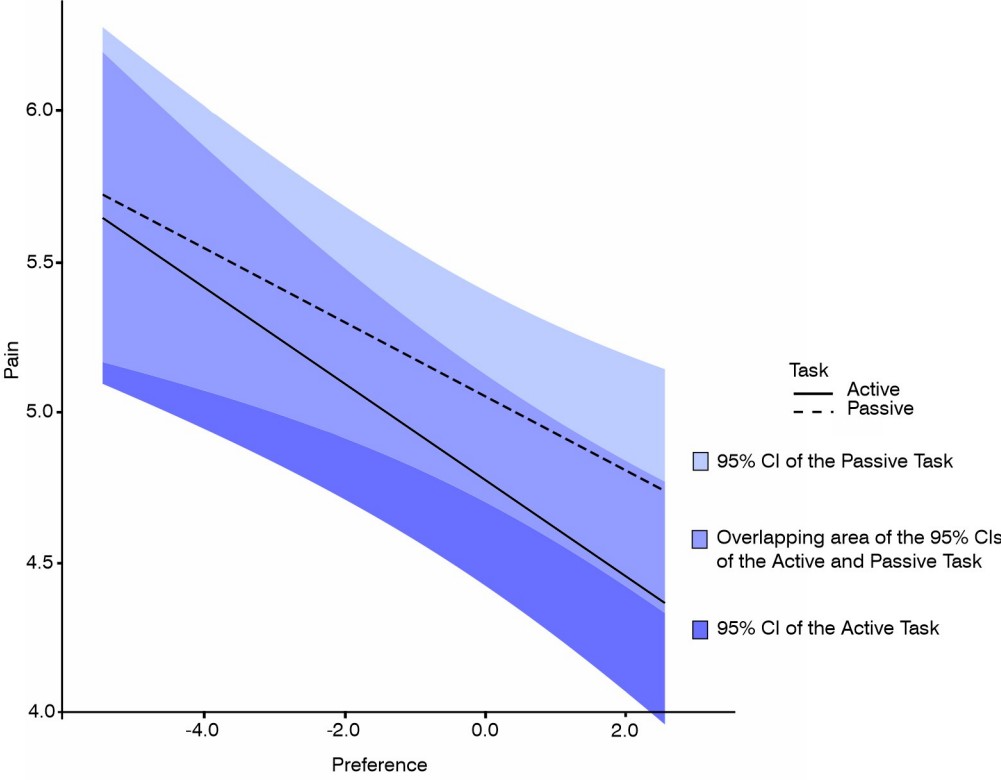

**Fig 4. Preference effect on the perceived pain for the music trials.** Data are shown separately for the active tapping task and the passive control task. Perceived pain (rated on a scale ranging from 1 to 9) was reduced with increasing preference (rated on a scale ranging from 1 to 9 and mean-centered) and while performing the active tapping task (solid black line) compared to the passive control task (dashed black line). Blue shaded areas represent the respective 95% Confidence Intervals (CIs) and their overlapping area.

not significant (β = -0.04, *SE* = 0.05, *p* = .453) and notably, no effect of the covariate *Familiarity* was observed (β = 0.03, *SE* = 0.02, *p* = .222). For all inferential statistics see S5 Table.

The second LME analysis on involved emotional mechanisms examined the effect of *Condition* and *Task* on the felt pleasantness. Results revealed a significant main effect of *Condition* (β = 1.02, *SE* = 0.25, *p* = .009), reflecting that felt pleasantness was higher while listening to music (compared to silence; see red vs. blue violin plots in Fig 5). In addition, this LME analysis revealed a trend for an interaction of *Condition* and *Task* (*p* = .073), indicating that the felt pleasantness of the active tapping task was descriptively larger while listening to music (*mean difference* = 0.18 scale points, *SE* = 0.08, *d* = 0.30) than during silence (*mean difference* = 0.06 scale points, *SE* = 0.11, *d* = -0.07). The difference in felt pleasantness between music with tapping vs. silence without tapping (*mean difference* = 1.08 scale points, *SE* = 0.16, see dashed grey horizontal line in Fig 5) had a large effect size [32] of *d* = 0.86. The main effect of *Task* was not significant (β = 0.06, *SE* = 0.07, *p* = .391). For all inferential statistics see S6 Table. A descriptive analysis of arousal ratings is provided in S7 Table.

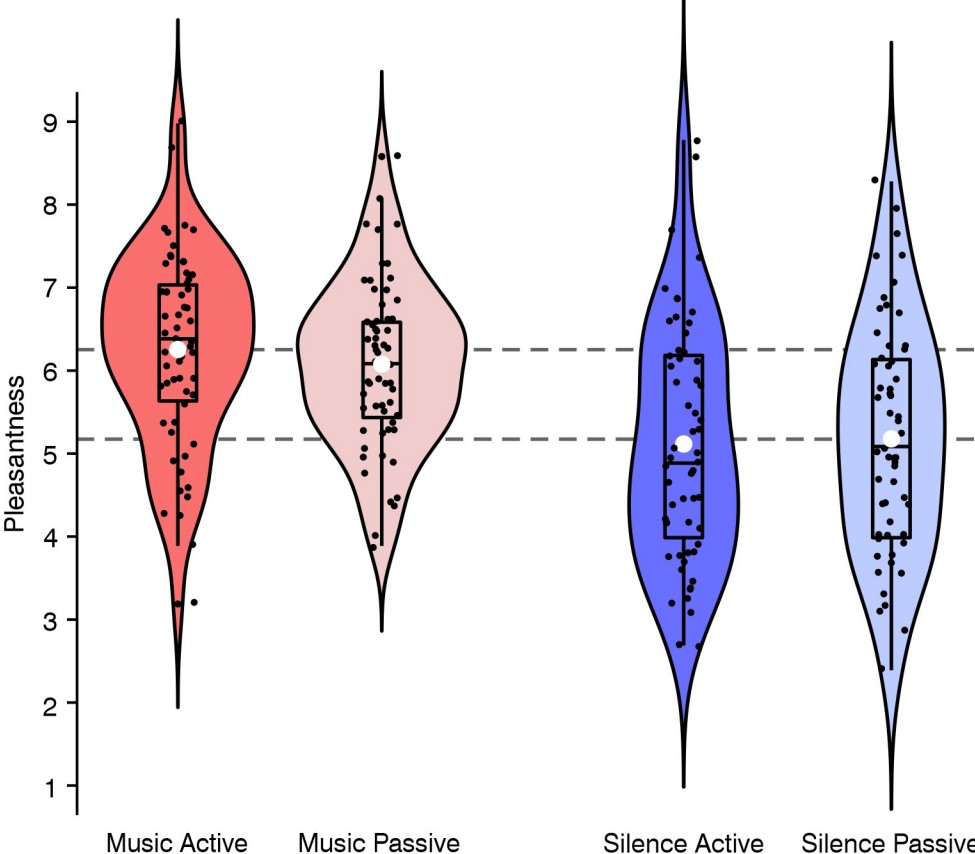

**Fig 5. Within-subjects felt pleasantness per experimental condition.** The violin plots show the distribution density of the felt pleasantness (rated on a scale ranging from 1 to 9) during music (red) or silence (blue) while performing either an active tapping task (darker colors) or a passive control task (lighter colors). Note that pleasantness ratings were descriptively highest for trials in which participants were tapping to the music. The embedded box-and-whisker plots represent the 25th and the 75th percentiles of the distributions, respectively. Upper and lower whiskers extend from the hinge to the largest/smallest value no further than 1.5 * inter-quartile range. The vertical lines in the boxes indicate median values, and the white disks indicate the means. The black dots show the jittered data points, and the dashed grey horizontal lines in the background represent the mean difference of the felt pleasantness between music with tapping and silence without tapping.

## Discussion

Our results show that participants felt less pain while actively tapping to music, compared with merely listening to music. This finding indicates that sensorimotor synchronization to music has analgesic effects. Compared to music listening without tapping, tapping to music elicited a moderate pain-reducing effect, and compared to the silent control condition without tapping, tapping to music elicited a large, clinically significant [32] pain-reducing effect ($d$ = 0.93). Our observation of a reduction of perceived pain while passively listening to music (compared with silence) is well in accordance with previously reported pain-reducing effects of music listening in clinical settings [1–4]. Likewise, our finding of a significant reduction of perceived pain while actively tapping (compared to the passive control task, independent of whether music or silence was presented) is consistent with the well-established analgesic effects of distracting attention from a painful stimulus [27–30]. The inclusion of the mean-centered Edinburgh Handedness Inventory index [EHI] [34] as a covariate in the LME analysis did not affect the referential pattern, and the effect of the predictor was not significant (see S4 File). The same pattern of results was observed when including the categorical fixed-effect factor Gender as a covariate in the LME analysis (see S5 File).

Notably, our findings are consistent with the hypothesis that sensorimotor synchronization to music reduces pain by virtue of EOS activation [12], thus supporting psychological accounts on the effects of movement synchronization to music on social bonding [11, 12, 16–18], trust [45, 46], and cooperation [47], as well as evolutionary accounts on the adaptive utility of music due to the promotion of social bonding by coordinated, synchronized musical activity [17].

We also investigated potential mechanisms underlying the analgesic effects of sensorimotor synchronization to music. So far, the (causal) mechanisms underlying the pain-reducing effects of (passive) music listening are not fully understood, but previous work has implicated attentional and emotional mechanisms as the two main candidates [48–50]. Our study was purposefully set up in a way that both music perception (compared to silence) and the active tapping task (compared to the passive control task) likely distracted participants' attention away from the pain stimulus [51]. Hence, tapping to music would have the strongest attention-capturing effect, resulting in the largest pain-reducing effects. However, the finding that tapping in silence did not significantly reduce pain (compared with silence without tapping, see also S4 Table) indicates that the attentional distraction by the tapping cannot alone explain the pain reducing effects of tapping to music.

Additionally, since pain-reduction is commonly used as a proxy of EOS activation [11–13, 16–20] and EOS activation has well-documented effects on analgesia and pleasure [24, 52], emotional mechanisms are likely involved in the analgesic effects of sensorimotor synchronization to music. To test this assumption, we analyzed participants' ratings of their emotional state in terms of their felt pleasantness, preference for the music, and familiarity with the music. Familiarity was included in this analysis due to a possible association between familiarity of music and (musical) preference, e.g it had been reported that the (dopaminergic) reward network is activated more strongly by familiar than unfamiliar music [26, 53, 54]. Results showed that perceived pain was significantly reduced when listening to strongly preferred music (unaffected by familiarity), consistent with previous research on preference in the context of pain-reducing effects of music (e.g., participants tolerate painful stimuli significantly longer when listening to their own preferred music [55–58]). Consistent with our findings, a recent study on real life acute pain showed that personal choice over music (cognitive agency) is a strong predictor for successful music-induced analgesia [59]. Reinforcing these findings, another recent study showed that listening to liked music during acute pain significantly lowered pain ratings compared to disliked music or no music and

decreased brain activity in pain-related areas compared to disliked music [60], In our study, participants did not choose the songs themselves, but it would be natural to assume that highly preferred songs would have been selected had participants had the choice. Moreover, our findings indicate that tapping to music was perceived as more pleasant than merely listening to music, suggesting that it was more rewarding to tap to the music, also consistent with a stronger EOS activation.

## Limitations

Our behavioral approach only provides an indirect measure of the activation of the EOS. This limitation warrants further discussion because the mixed-trial design of our study poses the question as to whether the level of endogenous opioids can vary every minute, forty times in a row, without contaminating the subsequent trial. We believe that our results are physiologically plausible, because in a previous functional magnetic resonance imaging (fMRI) study, also using a mixed-trial design, we showed that 30 seconds of pleasant vs. unpleasant music (corresponding to the length of the musical stimuli used in the present study) are sufficient to elicit activity changes in the hippocampus [61]. The hippocampus has receptors for endorphins (i.e., brain-produced opioids), and the hippocampus can produce opioids and release them into different regions of the brain, e.g., into the reward system [62]. Moreover, the hippocampus can trigger the release of endorphins in the hypothalamus (a major source of endogenous opioids) [62]. Thus, it is reasonable to assume that, in our previous fMRI study, the flexible activation and deactivation of the hippocampus, which occurred as a function of the (un)pleasantness of the music, was at least in part associated with EOS system activity. A similar time-course of EOS system activity might have been at work in the present study. However, future studies might verify our approach by directly measuring EOS activity, e.g., with positron emission tomography.

Another limitation of our study is that our four experimental conditions (especially music with tapping vs. silence with tapping) might have differed in their task difficulty. Thus, we cannot exclude the possibility that our results are confounded by effects of task difficulty. However, our results indicate that pleasantness and familiarity contributed to the modulation of pain (see above), and therefore the pain-reducing effects of sensorimotor synchronization to music cannot be explained by task difficulty only. Nevertheless, it remains a matter of future studies to elucidate the influence of task difficulty on pain-reducing effects of sensorimotor synchronization to music.

## Conclusion

Results demonstrate that sensorimotor synchronization to music significantly impacts pain perception and can amplify the well-established pain-reducing effect of merely listening to music. Our results shed new light on the mechanisms underlying pain reduction with music, suggesting that such effects are driven by social bonding, attention, emotion, and preference. Our findings support the recent hypothesis that social bonding emerges with music due to the activation of the endogenous opioid system (EOS) with sensorimotor synchronization [12], but future studies might apply more direct measures of EOS activity, of social bonding, and of task difficulty in trials with and without sensorimotor synchronization. Our discovery of pain-reducing effects of sensorimotor synchronization to music can easily be applied in a wide range of settings, such as clinical settings (using music as an adjunct treatment of pain) [63], in music therapy (e.g. for chronic pain) [64], and in everyday life (for the management of acute pain).

## Supporting information

**S1 Protocol.**
(DOCX)

**S2 Protocol.**
(DOCX)

**S3 Protocol.**
(DOCX)

**S1 Checklist. CONSORT checklist.**
(DOC)

**S1 Table. Music excerpts played during the experiment.**
(DOCX)

**S2 Table. Descriptive statistics of the applied pressure and duration of the applied pressure.**
(DOCX)

**S3 Table. Inferential statistics of the LME analysis on single trial perceived pain.**
(DOCX)

**S4 Table. Paired-Samples t-tests (one-tailed) on the perceived pain between all four experimental conditions.**
(DOCX)

**S5 Table. Inferential statistics of the LME analysis on the single trial perceived pain for music trials.**
(DOCX)

**S6 Table. Inferential statistics of the LME analysis on the single trial felt pleasantness.**
(DOCX)

**S7 Table. Descriptive statistics of the felt arousal per experimental condition.**
(DOCX)

**S1 File. Supplementary methods for the pressure data.**
(DOCX)

**S2 File. Experimental instructions.**
(DOCX)

**S3 File. Supplementary methods: Experimental design and data analysis on emotional mechanisms underlying the pain-reducing effect of sensorimotor synchronization to music.**
(DOCX)

**S4 File. Data analysis & results on the pain-reducing effect of sensorimotor synchronization to music including the mean-centered EHI as a covariate to test for confounding effects of handedness.**
(DOCX)

**S5 File. Data analysis & results on the pain-reducing effect of sensorimotor synchronization to music including gender as a covariate to test for confounding effects.**
(DOCX)

## Author Contributions

**Conceptualization:** Stefan Koelsch.

**Formal analysis:** Lucy M. Werner, Stavros Skouras, Laura Bechtold.

**Investigation:** Lucy M. Werner.

**Methodology:** Stavros Skouras, Ståle Pallesen, Stefan Koelsch.

**Project administration:** Lucy M. Werner.

**Resources:** Ståle Pallesen, Stefan Koelsch.

**Software:** Stavros Skouras.

**Supervision:** Stefan Koelsch.

**Visualization:** Lucy M. Werner.

**Writing – original draft:** Lucy M. Werner.

**Writing – review & editing:** Stavros Skouras, Laura Bechtold, Ståle Pallesen, Stefan Koelsch.

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
