## [Decision Letter · Decision Letter 0]

20 Mar 2023

PONE-D-23-00769Sensorimotor synchronization to music reduces painPLOS ONE

Dear Dr. Koelsch,

Thank you for submitting your manuscript to PLOS ONE. After careful consideration, we feel that it has merit but does not fully meet PLOS ONE’s publication criteria as it currently stands. Therefore, we invite you to submit a revised version of the manuscript that addresses the points raised during the review process.

The reviewers suggest that you provide more information about sample characteristics and about how emotions were specifically measured in the study. They recommend that you discuss other options for active engagement with music and health-related effects, particularly in the area of music therapy. In addition, the statistical analysis needs to be revised to be more specific about the analyses performed for the targeted endpoints. Please find the detailed comments appended below.

We look forward to receiving your revised manuscript.

Kind regards,

Alex Schaefer, PhD

Associate Editor

PLOS ONE

Journal Requirements:

3. We note that Figure 2 in your submission contain copyrighted images. All PLOS content is published under the Creative Commons Attribution License (CC BY 4.0), which means that the manuscript, images, and Supporting Information files will be freely available online, and any third party is permitted to access, download, copy, distribute, and use these materials in any way, even commercially, with proper attribution. For more information, see our copyright guidelines: http://journals.plos.org/plosone/s/licenses-and-copyright.

Reviewers' comments:

Reviewer's Responses to Questions

**Comments to the Author**

1. Is the manuscript technically sound, and do the data support the conclusions?

Reviewer #1: Yes

Reviewer #2: Partly

Reviewer #3: Yes

2. Has the statistical analysis been performed appropriately and rigorously? 

Reviewer #1: Yes

Reviewer #2: No

Reviewer #3: Yes

3. Have the authors made all data underlying the findings in their manuscript fully available?

Reviewer #1: Yes

Reviewer #2: Yes

Reviewer #3: Yes

4. Is the manuscript presented in an intelligible fashion and written in standard English?

Reviewer #1: Yes

Reviewer #2: Yes

Reviewer #3: Yes

5. Review Comments to the Author

Reviewer #1: This is an interesting study examining the roles of tapping and listening to music on pain perception (2 x 2 design, within-subjects manipulation). The paper is very well written, the study has been conducted rigorously, and the methods are sound. The findings have relevant theoretical and clinical implications, and I look forward to seeing them published.

While I don’t have major concerns, I find the lack of Condition x Task interaction problematic for the argument that analgesic effects are specific to when we synchronize with music (which seems to be the take-home message of the paper). Simple comparisons suggest that this is the case, but without the interaction the statistical support is rather weak. Describing a p = .140 (one-sided) as trend-level is hardly acceptable. I therefore suggest that the authors tone their argument down and change the wording used to describe the interaction.

Reviewer #2: This is basically a cross sectional design, that is to say an experiment using a 2x2 design with the within-subject factors Condition (music, silence) and Task (active, passive), resulting in four experimental trial types. The analysis is basic to this type of descriptive format. The sample size is historically justified at about 59 subjects with an age range of about 19 to 35 and a targeted effect size of about 0.4 for the endpoints of interest. What is a naïve random sample in this context?

The investigators mention 40 experimental trials and a basic alpha of 0.05. However, one wonders what adjustment is made in the type I error to accommodate all these trials. Also, the sample is not well described. Is there a gender breakdown and what characteristics of these individuals would allow for any relevant subset breakdowns if there were underlying characteristics of the population of interest that might be statistically associated with the outcomes? The results are mainly t-tests, violin or box plots with some regression considerations.

What is the actual physical setting of this experiment? Is it a clinical setting as one might expect?

When discussing the Pressure algometry the investigators make statements such as, “The statistical analysis of these data showed that neither the applied pressure, nor the duration of applied pressure, differed significantly for the within-subjects factors Condition (music, silence; pressure intensity: F(1,58) = 0.33, p = .568; pressure duration: F(1,58) = 1.62, p = .209), nor Task (active, passive; pressure intensity: F(1,58) = 0.33, p = .568; pressure duration: F(1,58) = 0.66, p = .419). Likewise, no significant interaction etc.” What exactly is the statistical analysis being performed? Presumably a multiway ANOVA? The manuscript could use a good rewrite being more specific on the analyses being conducted for the targeted endpoints.

Reviewer #3: This is an interesting and relevant paper that contributes to better understanding the potential of music to reduce pain. In essence, the authors provide evidence that sensorimotor synchronization with listening to music increases the beneficial effects of music, using a laboratory pain induction protocol in healthy volunteers. The following limitations should be addressed in a potential revision of the manuscript:

1. The authors introduce the EOS as a major mechanism that underlies the beneficial effect of music on pain. The EOS is discussed quite prominently, being mentioned in the abstract, and being introduced in the theory section. However, the study does not measure the EOS directly, so all text related to EOS remains mere speculation. As such, discussing the EOS as a potential mechanism that might mediate the effects observed in this study should be moved to the discussion. The authors already devote a part of their discussion to the EOS, but they should remove the EOS from the intro (and from the abstract), as the reader will expect that the EOS is tested although it´s not.

2. Sensorimotor synchronization is just one possibility of active engagement in/with music. It would be of interest if the authors would discuss other options of active engagement in music and health-related effects, particularly in the area of music therapy, and why they selected sensorimotor synchronization out of many other possibilities.

3. The introduction contains a major part that belongs to the methods section. Starting with line 73, in which the consort graph is introduced, to line 103, the entire text is devoted to describing details of the protocol and the assessments. This needs to be removed from the intro.

4. The problem then is that, without the part on the EOS and without the methods part, the introduction is not more than 1 page in the end. While there is beauty in a short introduction, the authors might want to consider including a more comprehensive summary of the available literature on the relationship between music and pain, and discuss in more detail the effects of attention on pain (which is now only being mentioned very briefly, but clearly attention/distraction from pain will play a huge role). Also see comment number 6.

5. The authors should provide more information regarding some sample characteristics:

o They examined an equal number of women and men. It looks like they had a hypothesis about potential gender effects. Did they consider including gender as a co-variable? Did they consider testing potential gender differences?

o Why this exact number of participants? The power analysis is generic; ideally, the reasoning for a certain effect size should be based on theoretical assumptions and/or already available empirical evidence. This needs to be elaborated.

6. There is a potential conceptual issue with how emotions were measured. Variables such as preference for music, familiarity with music, or perceived pleasantness within music (i.e. not felt pleasantness) are not considered emotions. The authors should introduce what they consider emotions and measure those – or use another term for these key variables. In relation to issue number 4, the authors could elaborate on this important aspect of their study in the introduction (because right now, there is no actual justification for the inclusion and relevance of these variables).

6. PLOS authors have the option to publish the peer review history of their article (what does this mean?). If published, this will include your full peer review and any attached files.

Reviewer #1: No

Reviewer #2: No

Reviewer #3: **Yes: **Urs Nater

---

## [Author Response · Author response to Decision Letter 0]

1 May 2023

“Sensorimotor Synchronization to Pain” - Point-by-point response to reviewers (also available as word document "Response to Reviewers.docx")

Editor comment:

The reviewers suggest that you provide more information about sample characteristics and about how emotions were specifically measured in the study. They recommend that you discuss other options for active engagement with music and health-related effects, particularly in the area of music therapy. In addition, the statistical analysis needs to be revised to be more specific about the analyses performed for the targeted endpoints. Please find the detailed comments appended below.

Response: We would like to express our gratitude to the editor for providing us with valuable feedback and suggestions regarding our manuscript. We took great care in reviewing and considering all critical issues that were brought to our attention. We want to highlight that we fully agree with all comments and have addressed each one separately in the following.

Reviewer Comments:

Reviewer 1

This is an interesting study examining the roles of tapping and listening to music on pain perception (2 x 2 design, within-subjects manipulation). The paper is very well written, the study has been conducted rigorously, and the methods are sound. The findings have relevant theoretical and clinical implications, and I look forward to seeing them published.

R1.1. While I don’t have major concerns, I find the lack of Condition x Task interaction problematic for the argument that analgesic effects are specific to when we synchronize with music (which seems to be the take-home message of the paper). Simple comparisons suggest that this is the case, but without the interaction the statistical support is rather weak. Describing a p = .140 (one-sided) as trend-level is hardly acceptable. I therefore suggest that the authors tone their argument down and change the wording used to describe the interaction.

Response: Thank you very much, we appreciate your comment. We agree that referring to a p-value of .140 (one-sided) as trend-level is not acceptable. Therefore, we removed this sentence from our manuscript. Nevertheless, we would like to emphasize that the absence of a significant interaction does not necessarily mean that the tapping was not effective. 

Supporting that, we would like to cite Page (2014): “From a clinical perspective, the presence (or absence) of statistically significant differences is of limited value. In fact, a non-significant outcome does not automatically imply the treatment was not clinically effective because small sample sizes and measurement variability can influence statistical results.1 Other factors, such as treatment effect calculations and confidence intervals offer much more information for clinicians to assess regarding the application of research finding, including both the magnitude and direction of a treatment outcome [1].

Incorporating the above-mentioned reasoning by Page [1], we attempted to tone down our argument and changed the wording to describe the interaction. 

For example, we now write:

“Despite the observation of the strongest pain-reducing effect of tapping to music in the descriptive analysis, the LME analysis failed to reveal a significant interaction. However, it is important to note that the absence of a significant interaction does not necessarily imply that the tapping was not effective. From a clinical perspective, the absence of statistically significant differences must be carefully scrutinized [43, 44]. Given the clinical relevance of our effects, and the fact that the results pattern nominally followed our a priori hypotheses, we separately examined and statistically compared the conditions. On that account, we conducted a simple slope analysis with Task as predictor and Condition as moderator. This simple slope analysis revealed a significant effect of the Task only while listening to music (β = -0.29, SE = 0.10, p = .005) but not during silence (β = -0.12, SE = 0.10, p = .225). Consequently, the difference in perceived pain was most pronounced between music with tapping vs. silence without tapping (mean difference = 0.93 scale points, SE = 0.13, see dashed grey horizontal lines in Fig 3), with a large effect size of d = 0.93 (effect sizes are interpreted according to Cohen’s convention [32]). This effect was also confirmed by a paired-samples t-test, t(58) = -7.17, one-sided p < .001 (for paired-samples t-tests on the perceived pain between all four conditions see S3 Table).”

Reviewer 2

R2.1. This is basically a cross sectional design, that is to say an experiment using a 2x2 design with the within-subject factors Condition (music, silence) and Task (active, passive), resulting in four experimental trial types. The analysis is basic to this type of descriptive format. The sample size is historically justified at about 59 subjects with an age range of about 19 to 35 and a targeted effect size of about 0.4 for the endpoints of interest. What is a naïve random sample in this context?

Response: Thank you, naïve in this context was supposed to mean naïve with regard to the hypotheses, which we now formulated accordingly in the revised manuscript. 

R2.2. The investigators mention 40 experimental trials and a basic alpha of 0.05. However, one wonders what adjustment is made in the type I error to accommodate all these trials. Also, the sample is not well described. Is there a gender breakdown and what characteristics of these individuals would allow for any relevant subset breakdowns if there were underlying characteristics of the population of interest that might be statistically associated with the outcomes? The results are mainly t-tests, violin or box plots with some regression considerations.

Response: Thank you for this valuable comment. The applied analysis does not include trial-wise inferential testing. In contrast, the p-value with the stated alpha level is estimated using the Satterthwaite method based on sample sizes and standard deviations once after the beta was estimated based on the single trial data. Thus, the type I error is unaffected by the amount of trials. 

A gender breakdown was not included in the previous version of the manuscript. However, the comment made us realize that it might be interesting for readers to know what influence gender plays for our results, so we recalculated our main analysis including gender as a covariate to test for potential confounding effects and added this information to the revised manuscript. The results of this analysis showed no significant effect of the covariate and no influence on our main findings. For details see Supplementary Information (S5 File). We believe that there are otherwise no characteristics for meaningful and theory-driven breakdowns in our data, as our sample was very homogeneous e.g in terms of age and education level. 

R2.3. What is the actual physical setting of this experiment? Is it a clinical setting as one might expect?

Response: The investigation was carried out in a non-clinical laboratory setting at the Department of Biological and Medical Psychology at the University of Bergen, as the sample only included healthy participants. We added this information to the revised manuscript. 

R2.4. When discussing the Pressure algometry the investigators make statements such as, “The statistical analysis of these data showed that neither the applied pressure, nor the duration of applied pressure, differed significantly for the within-subjects factors Condition (music, silence; pressure intensity: F(1,58) = 0.33, p = .568; pressure duration: F(1,58) = 1.62, p = .209), nor Task (active, passive; pressure intensity: F(1,58) = 0.33, p = .568; pressure duration: F(1,58) = 0.66, p = .419). Likewise, no significant interaction etc.” What exactly is the statistical analysis being performed? Presumably a multiway ANOVA? The manuscript could use a good rewrite being more specific on the analyses being conducted for the targeted endpoints.

Response: We would like to thank the reviewer for this comment. We did not want to write in detail in the manuscript the statistical analysis we performed here, as it was not part of the main analysis, and this seemed to be too confusing. Nevertheless, your comment made us realize that we need to provide at least information on what statistical analysis was performed here. We added to our manuscript that we performed repeated measures ANOVAs (for details, see S1file and S2 Table, respectively). 

Reviewer 3

This is an interesting and relevant paper that contributes to better understanding the potential of music to reduce pain. In essence, the authors provide evidence that sensorimotor synchronization with listening to music increases the beneficial effects of music, using a laboratory pain induction protocol in healthy volunteers. The following limitations should be addressed in a potential revision of the manuscript: 

R3.1. The authors introduce the EOS as a major mechanism that underlies the beneficial effect of music on pain. The EOS is discussed quite prominently, being mentioned in the abstract, and being introduced in the theory section. However, the study does not measure the EOS directly, so all text related to EOS remains mere speculation. As such, discussing the EOS as a potential mechanism that might mediate the effects observed in this study should be moved to the discussion. The authors already devote a part of their discussion to the EOS, but they should remove the EOS from the intro (and from the abstract), as the reader will expect that the EOS is tested although it´s not. 

Response: We would like to thank the reviewer for this comment, which helped us realize that our wording in the introduction regarding the EOS was misleading. As a result, we have rephrased this section to clarify that we are testing the hypothesis of pain reduction through sensorimotor synchronization to music, rather than the hypothesis of EOS activation. However, it is worth noting that it is common to interpret pain reduction as a proxy for EOS activation (see [2-9]). Therefore, we only refer to EOS activation in the abstract as a motivating factor for our study. In addition, in our limitations section we emphasize that our behavioral approach only provides an indirect measure of the activation of the EOS. 

R3.2. Sensorimotor synchronization is just one possibility of active engagement in/with music. It would be of interest if the authors would discuss other options of active engagement in music and health-related effects, particularly in the area of music therapy, and why they selected sensorimotor synchronization out of many other possibilities.

Response: We would like to thank the reviewer for this comment. Music has many therapeutic effects, and especially in music therapy, social interaction and synchronized movement to a musical pulse in a group are often essential. Our intentional decision was to study the impact of sensorimotor synchronization to music on pain perception in isolation. This was done to gain a deeper understanding of one of the most widely studied and most replicable effects of music in clinical settings - pain-reducing effects of music listening.

R3.3. The introduction contains a major part that belongs to the methods section. Starting with line 73, in which the consort graph is introduced, to line 103, the entire text is devoted to describing details of the protocol and the assessments. This needs to be removed from the intro.

Response: We would like to thank the reviewer for this comment. We agree that the introduction was too detailed in describing the protocol and assessments, especially since this information is already covered in the figure legend of Figure 2. However, we did not completely remove this section from the introduction. Instead, we shortened it, as we believe it is essential to briefly introduce the conceptual operationalization and the study design in order, to derive our hypotheses in the following. The Consort Flow Diagram is located here because, according to the journal's guidelines, it has to be the first figure in the manuscript.

R3.4. The problem then is that, without the part on the EOS and without the methods part, the introduction is not more than 1 page in the end. While there is beauty in a short introduction, the authors might want to consider including a more comprehensive summary of the available literature on the relationship between music and pain, and discuss in more detail the effects of attention on pain (which is now only being mentioned very briefly, but clearly attention/distraction from pain will play a huge role). Also see comment number 6.

Response: Thank you for this comment. As mentioned in our response to comment R3.1. and R3.3, we agree, that our parts on the EOS in the introduction and the parts on the protocol and assessments were either misleading or too long. Therefore, we revised and condensed these parts accordingly. As stated in our response to comment R3.2., it was our intentional decision to study the impact of sensorimotor synchronization to music on pain perception in isolation. We have intentionally chosen to maintain this focus, and thus did not extensively discuss the general relationship between music and pain at this point and will report on the effects of attention on pain perception in the discussion section.

R3.5. The authors should provide more information regarding some sample characteristics:

R3.5.1 They examined an equal number of women and men. It looks like they had a hypothesis about potential gender effects. Did they consider including gender as a co-variable? Did they consider testing potential gender differences?

Response: We would like to thank the reviewer for this comment. Gender effects were not a specific focus of our study, but we strived to maintain a balanced distribution of gender in our sample. During the revision process, we re-analyzed our main results with gender included as a covariate to investigate potential confounding effects. This analysis was not previously reported in the manuscript but is now included in the revised version. For details see comment R2.2 and Supplementary Information (S5 File). 

R3.5.2 Why this exact number of participants? The power analysis is generic; ideally, the reasoning for a certain effect size should be based on theoretical assumptions and/or already available empirical evidence. This needs to be elaborated.

Response: Thank you very much for this comment and for bringing to our attention that our reported effect size is only generic. We apologize for any confusion caused by our error in referencing the wrong source here and corrected this mistake in the revised manuscript. The derivation of our effect size was based on the results of the two meta-analyses by Cepeda et al. [10] and Lee [11]. Additionally, we have slightly rephrased this paragraph to enhance clarity: 

“This sample size was based on an a-priori sample size estimation using G*Power (version 3.1.9.6) [31] which indicated that for the detection of differences between listening and not listening to music with an α-level of 0.05 (one-tailed) and a statistical power of 90% assuming a small to medium effect size of Cohen’s d = 0.4 [32] based on Cepeda et al. [1] and Lee [3], a sample of n = 59 is adequate.”

R3.6. There is a potential conceptual issue with how emotions were measured. Variables such as preference for music, familiarity with music, or perceived pleasantness within music (i.e. not felt pleasantness) are not considered emotions. The authors should introduce what they consider emotions and measure those – or use another term for these key variables. In relation to issue number 4, the authors could elaborate on this important aspect of their study in the introduction (because right now, there is no actual justification for the inclusion and relevance of these variables).

Response: We would like to express our gratitude to the reviewer for this comment, which has highlighted a potential misunderstanding regarding the reporting of our emotion variables. We aimed to investigate the participants' subjective experiences of "felt pleasantness" and "felt arousal". To improve clarity, we have rephrased these terms accordingly. We also included in our revised manuscript that the familiarity and preference ratings were measured not only to investigate their potential contributions to pain reduction, but also to cover additional aspects with emotional implications.

 

References 

1. Page P. Beyond statistical significance: clinical interpretation of rehabilitation research literature. Int J Sports Phys Ther. 2014;9(5):726–36.

2. Granot R. Music, pleasure, and social affiliation: Hormones and neurotransmitters. The Routledge Companion to Music Cognition: Routledge; 2017. p. 101-11.

3. Tarr B, Launay J, Dunbar RI. Music and social bonding:“self-other” merging and neurohormonal mechanisms. Front Psychol. 2014;5:1096.

4. Dunbar RI, Kaskatis K, MacDonald I, Barra V. Performance of music elevates pain threshold and positive affect: implications for the evolutionary function of music. Evol Psychol. 2012;10(4):147470491201000403.

5. Launay J, Tarr B, Dunbar RIM. Synchrony as an Adaptive Mechanism for Large-Scale Human Social Bonding. Ethology. 2016;122(10):779-89. doi: 10.1111/eth.12528. PubMed PMID: WOS:000383343100001.

6. Savage PE, Loui P, Tarr B, Schachner A, Glowacki L, Mithen S, et al. Music as a coevolved system for social bonding. Behav Brain Sci. 2020:1-36.

7. Tarr B, Launay J, Dunbar RI. Silent disco: dancing in synchrony leads to elevated pain thresholds and social closeness. Evol Hum Behav. 2016;37(5):343-9. Epub 2016/08/20. doi: 10.1016/j.evolhumbehav.2016.02.004. PubMed PMID: 27540276; PubMed Central PMCID: PMCPMC4985033.

8. Cohen EE, Ejsmond-Frey R, Knight N, Dunbar RI. Rowers' high: behavioural synchrony is correlated with elevated pain thresholds. Biol Lett. 2010;6(1):106-8.

9. Fritz TH, Bowling DL, Contier O, Grant J, Schneider L, Lederer A, et al. Musical Agency during Physical Exercise Decreases Pain. Frontiers in Psychology. 2018;8. doi: ARTN 231210.3389/fpsyg.2017.02312. PubMed PMID: WOS:000422691400001.

10. Cepeda MS, Carr DB, Lau J, Alvarez H. Music for pain relief. Cochrane Database Syst Rev. 2006;(2).

11. Lee JH. The effects of music on pain: a meta-analysis. Journal of music therapy. 2016;53(4):430-77.

---

## [Decision Letter · Decision Letter 1]

6 Jun 2023

PONE-D-23-00769R1Sensorimotor synchronization to music reduces painPLOS ONE

Dear Dr. Koelsch,

Thank you for submitting your revised manuscript to PLOS ONE. I have been asked to serve as Guest Editor for your manuscript and handle the revision process. Please note that I have previously served as Reviewer 3, and I was not involved in an editorial role for the initial submission. I believe you have addressed all of the previous issues that were raised by the three reviewers in an adequate manner. The publisher has also invited a statistical review, which you will find below. You will see that the statistical reviewer agrees with how you handled the reviewers´ requests, but asks you to provide some additional material as supplements. Therefore, we invite you to submit a revised version of the manuscript that includes updated supplementary material.

We look forward to receiving your revised manuscript.

Kind regards,

Urs M Nater

Guest Editor

PLOS ONE

Journal Requirements:

Reviewers' comments:

Reviewer's Responses to Questions

**Comments to the Author**

1. If the authors have adequately addressed your comments raised in a previous round of review and you feel that this manuscript is now acceptable for publication, you may indicate that here to bypass the “Comments to the Author” section, enter your conflict of interest statement in the “Confidential to Editor” section, and submit your "Accept" recommendation.

Reviewer #4: All comments have been addressed

2. Is the manuscript technically sound, and do the data support the conclusions?

Reviewer #4: Yes

3. Has the statistical analysis been performed appropriately and rigorously? 

Reviewer #4: Yes

4. Have the authors made all data underlying the findings in their manuscript fully available?

Reviewer #4: Yes

5. Is the manuscript presented in an intelligible fashion and written in standard English?

Reviewer #4: Yes

6. Review Comments to the Author

Reviewer #4: This review focusses on the statistical questions that have been raised previously, along with the authors' responses to the questions.

The main issues brought up in previous reviews concerned the potential for Type I error inflation due to the number of trials conducted with each subject, the lack of evidence for the interaction effect of Condition and Task in the primary analysis, and the lack of description of the other statistical analyses alluded to in the manuscript.

The first issue seems to have not been an actual problem as a linear mixed effects model was used which included random effects for both subject and musical excerpts. While one could definitely consider other model structures and assessments --- for example, testing for time effects within subject, or sequence effects --- this analysis should provide degrees of freedom that are at least reasonable.

Probably this question could have been avoided by including the actual analysis of variance tables for each analysis which would have allowed an assessment of the error degrees of freedom. This would still be useful to include as a supplementary file, since it would provide useful information for future researchers in estimating sample size or performing meta-analysis.

The authors have toned down the language around the main result in response to the second issue and have clarified that this somewhat limits the extent of the conclusions.

The other statistical analysis have been described at a high level now, dealing with the third primary issue. Again, it would be best to include the actual analysis of variance tables in the supplemental files. Additional analyses have also been performed within the same basic model structure, which has the benefit of providing a consistent framework for evaluating the results.

Since the data will be made available at publication, it will be possible for others to reevaluate the results using different analysis assumptions.

7. PLOS authors have the option to publish the peer review history of their article (what does this mean?). If published, this will include your full peer review and any attached files.

Reviewer #4: No

---

## [Author Response · Author response to Decision Letter 1]

6 Jul 2023

Thank you very much for the opportunity to revise our manuscript. We highly appreciate the constructive and helpful feedback from both the editors and the statistical review. We agree with all comments and have incorporated the advice to include the actual analysis of variance tables for the linear mixed effect analyses in the supplemental files of our revised manuscript.

---

## [Editor Report · Decision Letter 2]

17 Jul 2023

Sensorimotor synchronization to music reduces pain

PONE-D-23-00769R2

Dear Dr. Koelsch,

We’re pleased to inform you that your manuscript has been judged scientifically suitable for publication and will be formally accepted for publication once it meets all outstanding technical requirements.

Kind regards,

Urs M Nater

Guest Editor

PLOS ONE
---

## [Editor Report · Acceptance letter]

20 Jul 2023

PONE-D-23-00769R2 

Sensorimotor synchronization to music reduces pain 

Dear Dr. Koelsch:

I'm pleased to inform you that your manuscript has been deemed suitable for publication in PLOS ONE. Congratulations! Your manuscript is now with our production department. 

Kind regards, 

on behalf of

Dr. Urs M Nater 

Guest Editor

PLOS ONE